SENSH: a blockchain-based searchable encrypted data sharing scheme in smart healthcare

Luo Song 1
Tan Lihuan 1 tanlihuan0509@foxmail.com
Hu Tan 2
Hu Maoshuang 1
1 College of Computer Science and Engineering, Chongqing University of Technology , Chongqing , China
2 Key Laboratory of Public Big Data Security Technology, College of Mobile Communication , Chongqing , China
Alarcon-Aquino Vicente
Electronic publication date: 2025 Sep 8
Publication date: 2025
Volume: 11
Electronic Location ID: e3166
Received 2025 Apr 18; Accepted 2025 Aug 6
Copyright: © 2025 Luo et al.
Copyright year: 2025
Copyright holder: Luo et al.
License: This is an open access article distributed under the terms of the Creative Commons Attribution License, which permits unrestricted use, distribution, reproduction and adaptation in any medium and for any purpose provided that it is properly attributed. For attribution, the original author(s), title, publication source (PeerJ Computer Science) and either DOI or URL of the article must be cited.
License URL: https://creativecommons.org/licenses/by/4.0/

Keywords: Searchable encryption, Blockchain, Smart healthcare health management, Internet of Things (IOT), Data privacy protection

Funding: The authors received no funding for this work.

==============================
The rapid development of the Internet of Things technology has led to a boom in the adoption of intelligent healthcare management systems in the healthcare industry. However, it has also highlighted key issues such as security, privacy, and efficient query of medical data. Traditional methods for querying medical data suffer from severe data leakage risks, low query performance, and excessive storage space. This article proposes a comprehensive Secure ENcrypted Search for Health Scheme (SENSH) solution based on consortium blockchain and searchable encryption to address these challenges. SENSH enables efficient authorization management through Bloom filters, ensuring fast querying of large datasets by authorized users while saving storage space. It uses off-chain Advanced Encryption Standard (AES) and on-chain storage management for data protection, significantly reducing the likelihood of data exposure. The system is also enhanced with event triggering and logging mechanisms to support real-time monitoring and data tracing to meet audit compliance requirements. It provides version control and timestamping to accommodate dynamic data updates, employs an obfuscationfactor to prevent tag-based original data content leakage, and supports dynamic updating of tags to accommodate different access requirements. Experimental results show that SENSH excels in authorization management, privacy protection, defense against tampering, and anti-replay and Distributed Denial of Service (DDoS). Compared with existing schemes, SENSH has significant advantages in terms of gas consumption, computation cost, and execution time. It is particularly suited for the protection and efficient query of medical and health data.

Introduction

The smart healthcare system (SHS), a niche within the Internet of Things (IoT), is a data-centric framework for healthcare services. It combines multiple technologies, such as blockchain, artificial intelligence, cloud computing, and the Internet of Things to connect patient data, medical devices, and healthcare services (Bao, Qiu & Cheng, 2022) into an intelligent and convenient system. SHS aims to provide accurate health monitoring, comprehensive diagnostic services, integrated healthcare solutions, and preventive care. The types of data processed by SHS mainly include electronic medical records (EMR) (Li et al., 2021a), imaging data, and health monitoring data, which are large and complex. Real-time health monitoring data are collected from smart wearables (Guan et al., 2022) such as smart bracelets, smart patches, and smart shoes. As depicted in Fig. 1, IoT integrates these into the healthcare system, enabling dynamic health tracking and direct data integration into electronic healthcare systems. This integration deepens the convergence of health monitoring with medical records, helping healthcare organizations more accurately analyze patient health and adjust treatment plans and prevention strategies in real time. Even as the healthcare industry continues to integrate diverse technologies to drive development, the sharing and managing of healthcare data and privacy remain areas for improvement. In the context of big data, ensuring efficient data sharing and retrieval while maintaining privacy is a critical challenge (Yang et al., 2019) in intelligent healthcare.

Figure 1 Architecture of SHS with IOT.

Blockchain (Shen et al., 2024), a decentralized distributed ledger technology, ensures data immutability and transparency by encapsulating information into blocks and linking them chronologically to form a chain structure (Qu et al., 2024). However, the blockchain stores all information in a public ledger, which means anyone can view its contents, failing to meet the stringent requirements for data privacy protection in healthcare systems. Moreover, traditional search methods rely on retrieving and matching plaintext information to obtain the required data, an approach fraught with vulnerabilities. Since the data is not encrypted, it is highly susceptible to theft, copying, leakage, or tampering by external attacks or internal operators, directly exposing it to unauthorized access risks.

To address these issues, the integration of searchable encryption (SE) technology (Jiang et al., 2021) allows for the effective searching and querying of data even when it is encrypted, thus enabling the secure retrieval of sensitive data. Searchable encryption robustly safeguards data privacy and significantly mitigates the risk of data leakage. Therefore, by combining blockchain and searchable encryption technologies and leveraging their respective strengths, a secure, flexible, and controllable data management solution can be provided for intelligent medical and health management systems. This synergy promotes the dual primary objectives of secure sharing and efficient utilization of medical data (Wu et al., 2023).

The article proposes Secure ENcrypted Search for Health Scheme (SENSH), an asymmetric searchable encryption scheme for blockchain-based smart medical healthcare management systems. The scheme integrates data security, privacy protection, and efficient querying, solving the challenges of sensitive data leakage and difficult sharing. It provides a user rights management mechanism that ensures only authorized users can access and query data. Specifically, the system prevents unauthorized access and data leakage by authenticating through an authentication process that issues search tokens to data visitors. In the process of data storage and transmission, the scheme adopts the interaction mode of on-chain and off-chain collaboration. First, the patient’s publicizable plaintext information is placed on the cloud server to store the record; then, the symmetric Advanced Encryption Standard (AES) encryption algorithm is utilized under the chain to encrypt the sensitive plaintext information, and the generated ciphertext data is then re-encrypted by the cloud’s agent and stored to the cloud. This process ensures the integrity and immutability of sensitive data, while providing convenience for subsequent security queries. To enable searchability of ciphertext data, traditional methods often rely on dictionary collections for key-value pair queries. However, with exponential growth of data, the storage requirements of dictionary collections escalate, creating a significant bottleneck in terms of storage resources and computational efficiency. To address this problem, the system incorporates Bloom filters to improve storage and query efficiency. As an efficient data structure, the Bloom filter minimizes storage requirements while maintaining query accuracy by mapping encrypted keywords to bit arrays, effectively increasing retrieval efficiency and reducing computational costs. The optimized Bloom filter ensures compliance with privacy and security requirements while providing resource-efficient and effective query services.

The main contributions of this article are as follows: An asymmetric searchable encryption SENSH scheme for a blockchain-based smart medical health management system is proposed. The scheme integrates blockchain, searchable encryption, Bloom filters, and proxy re-encryption techniques. It ensures data immutability and transparency, enhances data privacy protection, reduces the risk of data leakage, and enhances the privacy of data sharing.

A searchable encryption scheme is proposed for on-chain and off-chain cooperative interaction. The off-chain data is protected by AES symmetric encryption and re-encryption, and ciphertext data is managed by on-chain storage mechanisms, reducing the risk of data leakage. The on-chain part implements tag encryption through asymmetric encryption, where the public key of the data owner encrypts the data tag. At the same time, the authorized user uses the private key for query operations. The scheme also supports tag data updating, which enhances the flexibility of data management to adapt to the access needs in different contexts.

Operation logs are recorded in real-time through the blockchain’s event-triggered function, ensuring event monitoring and traceability to meet the auditing requirements of healthcare systems. Applying Bloom filters, version control, and timestamps improves the efficiency of data query and storage utilization. It supports multi-version data tracking and management to ensure that the system adapts to the dynamic updating needs of medical data. In addition, introducing an obfuscationfactor mechanism improves query security and effectively prevents attackers from inferring data content, thus safeguarding data privacy and security management.

The efficiency and performance of the proposed scheme are evaluated through experiments, which mainly include storage efficiency, computation efficiency, and gas cost. The experimental results show that the proposed scheme can significantly improve the query efficiency and reduce the storage overhead under the premise of guaranteeing data privacy, and shows better performance in the aspects of authorization management, data generation and storage, and event logging, with better scalability and responsiveness.

The remainder of this article is structured as follows: ‘Related work’ reviews relevant work and current solutions. ‘Background’ describes the essential prerequisites for our proposed scheme, focusing on technologies like blockchain and searchable encryption that form its basis. The ‘System Overview’ provides an overview of the system architecture, including the framework and security model. The discussion of ‘Algorithm and Experimental Analysis’ discusses the scheme’s structure, main algorithms, experimental results, security, and performance analysis. The final section summarizes the work and suggests directions for future work.

Related work

In recent years, as the medical data digitization process has advanced and the demand for privacy protection has surged, the integration of blockchain and searchable encryption technology in healthcare has garnered significant interest. Searchable encryption (SE) enables keyword-based search over encrypted data without revealing sensitive content, and has become essential in protecting privacy in outsourced healthcare systems. A number of studies have explored the combination of blockchain and SE to support secure and traceable data sharing. However, most existing works focus on one or two aspects such as data immutability, encrypted storage, or basic access control, while overlooking critical requirements such as efficient on-chain authorization, fast encrypted query processing, and comprehensive attack resistance. These limitations motivate the development of more integrated and efficient frameworks like SENSH.

Azaria et al. (2016) introduced a decentralized electronic medical record management system leveraging blockchain for authentication, responsibility tracing, and data sharing, with a modular design and a proof-of-work mechanism that rewards participants for anonymized data provision, ensuring data security and network sustainability. However, direct blockchain storage of encrypted data might lead to storage and computational constraints as data volumes grow. Xia et al. (2017) presented the MeDShare system to address healthcare data sharing among big data custodians in a trustless environment, ensuring data security and traceability through cloud storage, but the decentralized nature of blockchain and data integrity verification increase system computation and storage needs, necessitating a privacy-data sharing balance. Nguyen et al. (2019) proposed a blockchain and interplanetary file system (IPFS)—an integrated framework for secure electronic health record (EHR) sharing on mobile cloud platforms. However, this combination raises cost concerns, and strict access control could impede emergency data access. Chase & Kamara (2010) developed a structured encryption model for distributed data storage. However, it needs help with querying efficiency amidst large labeled datasets, and direct chain storage fails to achieve accurate data distribution. Kamara & Papamanthou (2013) advanced a sub-linear, dynamically searchable encryption scheme using multi-core parallel computation and red-black-tree structures suitable for updating medical data. However, query efficiency still needs to improve in complex systems.

In Boneh et al. (2004) and Curtmola et al. (2006), separately exploring searchable public keys and symmetric encryption implements cryptographic data search operations. However, traditional search schemes risk privacy leakage, especially with multiple queries that might reveal sensitive patient information. Yue et al. (2016) proposed an HGD architecture with a smart contract and ICS for patient-controlled healthcare data sharing. However, it needs more support for dynamic privilege updates, failing to meet evolving real-world authorization needs. Chen et al. (2019) designed a ciphertext-based electronic surveillance recording and sharing system, constructing EHR indexes for privacy and efficient retrieval. However, it is limited to static data and cannot accommodate dynamic updates. Miao et al. (2021) introduced a multi-authority Multi-Authority Attribute-Based Keyword Search (MABKS) system to overcome performance bottlenecks in single-authority Ciphertext-Policy Attribute-Based Keyword Search (CP-ABKS). However, it lacks fine-grained access control and flexible queries, limiting privacy and scalability. The Blockchain-Based Search Attribute-Based Encryption (BC-SABE) scheme in Liu et al. (2020) addresses IoT data storage and access control with a decentralized critical management approach combining blockchain and attribute-based cryptography. However, it faces challenges with resource-constrained IoT devices, multi-user critical management complexity, and blockchain’s computational demands, showing significant limitations in dynamic updates and multi-keyword queries. Lastly, Mamta et al. (2021) combines Attribute-Based Searchable Encryption (ABSE) and blockchain consensus mechanisms for Cloud-Based Healthcare Cyber-Physical System (CCPS) to ensure data confidentiality and privacy, but high computational costs, especially with large data volumes, can impact system performance.

Li et al. (2021b) proposes a blockchain-based verifiable public key encryption scheme that utilizes the TrueBit network to outsource the verification task, ensures the validity of search results, and realizes fair payment between data owners and users through smart contracts. However, the scheme relies on the third-party verification nodes of TrueBit, which involves trust risks; meanwhile, the high gas consumption of smart contracts limits the efficiency of large-scale data scenarios, and the complexity of the fine-grained payment mechanism for multiple data owners is high, which increases the system overhead. In Ali et al. (2022), a hybrid deep neural network framework combining group theory and binary spring search algorithm is designed for intrusion detection and security search in the Internet of Medical Things (IIoT). The scheme guarantees secure storage and search of patient health records through blockchain and homomorphic encryption, and improves malicious attack detection with the help of deep neural networks. However, the scheme has high computational overhead, which makes it difficult to be deployed in resource-constrained IIoT devices; the algorithm complexity is high, which affects the emergency response in medical scenarios; and the flexibility of multi-keyword search and the support of dynamic data update are insufficient. Xiang & Zhao (2022) proposes a blockchain-based search attribute encryption scheme to safeguard the data integrity of e-health systems and realize fine-grained access control through blockchain. However, the performance of the scheme decreases significantly with the increase in policy complexity, which increases the storage and communication overhead and affects the scalability of the system. In Liu et al. (2023), a blockchain-based search agent signing and confidentiality scheme is designed to achieve centralized authorization management of medical data by patients using identity-based agent signatures while guaranteeing the integrity and traceability of search results. However, the scheme increases the operational burden on the user side, and the storage and verification overhead of the blockchain is large, making it difficult to operate efficiently in resource-constrained environments.

While considerable progress has been made in blockchain-based privacy protection and access control, few studies have provided an integrated solution that simultaneously addresses efficient on-chain authorization, privacy-preserving search, performance optimization, and attack resilience. This gap highlights the need for a lightweight yet comprehensive framework tailored to smart healthcare environments, which motivates the design of our solution presented in the following section.

Background

This section delves into the pivotal techniques for crafting an intelligent, searchable, and readily shareable healthcare management system. It harnesses blockchain-based asymmetric searchable encryption, augmented with Bloom filters, proxy re-encryption mechanisms, and symmetric encryption algorithms to facilitate efficient healthcare data management and secure data sharing. These technologies safeguard data security and privacy protection throughout data transmission and sharing.

Searchable encryption

Searchable encryption (SE) (Wang et al., 2022) is a cryptographic technique that empowers users to execute search operations on encrypted data without decryption. This technique permits data visitors to search using encrypted queries and match them against encrypted indexes or labels without decrypting the underlying data. The fundamental operational principle is as follows:

let Data represent the original data, En(Data) the encrypted form of the data, Label(En(Data)) the index or label associated with the encrypted data, Query the search operation, and En(Query) the encrypted query condition. Searchable encryption enables the computation of Query(En(Data), Label(En(Data))) to yield search results, all without decrypting the original data.

Searchable encryption technology facilitates efficient querying of encrypted data, thereby preserving data privacy. It addresses the limitation of traditional encryption algorithms that do not support direct querying of encrypted data. It is extensively utilized in cloud storage, distributed systems, and other environments demanding stringent data security. Depending on the encryption techniques and access control mechanisms, common types of searchable encryption include symmetric searchable encryption (SSE) (Li et al., 2024), asymmetric searchable encryption (ASE) (Niu et al., 2020), public key searchable encryption (PEKS) (Xu et al., 2018), and attribute-based searchable encryption (ABSE) (Liu et al., 2022). SSE is ideal for small-scale applications, where the data owner and user share a standard key, offering efficient query performance and minimal computational overhead. However, more is needed for multi-user scenarios. ASE enhances access control flexibility through public-key encryption and private-key querying, making it suitable for multi-user environments but with comparatively lower efficiency. PEKS encrypts data with the recipient’s public key, and the recipient uses the private key for querying, which is appropriate for multiparty sharing scenarios and offers heightened security, albeit with increased query efficiency and overhead. ABSE manages data access rights through user attributes, supporting fine-grained access control, and is well-suited for scenarios with intricate access requirements but incurs higher computational and storage costs. Consequently, selecting an appropriate searchable encryption scheme involves balancing security, query efficiency, and computational overhead against specific application needs.

Bloom filter

The Bloom filter, proposed by Burton Bloom in 1970 (Geravand & Ahmadi, 2013), is an efficient spatial data structure for quickly detecting whether an element belongs to a large data set (Dasgupta et al., 2018). It is realized by a bit array and a hash function; the bit array stores binary data, and the hash function maps elements to the bit array. When adding an element, the bit array is labeled by the hash function; when querying, the labeled bits are checked, and if all are 1, the element may exist; if 0 exists, the element does not exist. The Bloom filter ensures no false negatives, but false positives may exist. Its false positive rate depends on the size of the bit array and the number of hash functions, and a reasonable configuration can optimize the space and query efficiency. This structure is shown in Fig. 2. The key query initiated by the client is first probabilistically determined by the Bloom filter, and then the layered filtering logic significantly reduces the amount of redundant queries and improves the system throughput efficiency.

Figure 2 Bloom filter principle flowchart.

Proxy re-encryption

Proxy re-encryption (PRE) proposed by Zhou et al. (2023) and Tian et al. (2023) is a public-key cryptography that facilitates the transformation of encrypted data from one encryption key to another without revealing the underlying plaintext or decryption keys. This mechanism is critical to privacy because it allows a proxy to convert ciphertext originally encrypted under user A’s key to be readable under user B’s key using a re-encryption key, and the proxy remains unaware of the plaintext content.

As depicted in Fig. 3, the proxy does not require knowledge of the original data, ensuring that plaintext access is not compromised. In this schema, En_A and En_B represent the original ciphertext and the re-encrypted ciphertext, respectively, while K_A, K_B, and K_C are their corresponding private keys. K_R denotes the re-encryption key, and transforming one ciphertext to another is called the ReEncrypt operation.

Figure 3 Proxy re-encryption schematic.

In practical scenarios, cloud servers can serve as proxy entities, assisting blockchain systems in implementing decentralized access control. It allows data owners to dynamically manage data access rights by delegating the cloud server to carry out re-encryption operations and securely distribute encrypted data across the blockchain network, a solution particularly apt for medical applications. Consequently, data owners are relieved of the need to distribute the decryption key to each doctor; instead, they control access by empowering the cloud server to execute re-encryption tasks, safeguarding the privacy and security of shared data. With formidable computational resources, cloud servers can adeptly manage re-encryption computations, substantially minimizing operational delays, enhancing system performance, and bolstering security and privacy during data sharing.

AES symmetric encryption algorithm

Symmetric encryption, a foundational method in cryptography, utilizes the same key for encrypting and decrypting data. Compared to asymmetric encryption, symmetric encryption offers significantly higher computational efficiency, especially when dealing with large-scale medical records or IoT-generated data streams in smart healthcare environments. Among the well-known symmetric encryption algorithms are AES (Usman, Jan & He, 2017) and Data Encryption Standard (DES) (Tang et al., 2018), which has become the de facto standard in secure data protection. Our scheme opts for the AES algorithm due to its superior security and computational efficiency, which align with our system’s performance criteria. Specifically, we adopt AES-256, which employs a 256-bit key to provide strong resistance against brute-force attacks and enhanced confidentiality protection for sensitive health information. AES processes data in fixed-size blocks, each measuring 128 bits (or 16 bytes), and it accommodates key lengths of 128, 192, or 256 bits. Its compatibility with hardware acceleration further contributes to low-latency encryption on edge and cloud platforms, making it well-suited for use in smart healthcare systems. The encryption implementation principle is detailed as follows: (1) First, initialize the plaintext and divide it into 16 bytes for encryption.

(2) The initial transformation formula is as follows: (1) M[i,j]=M0⊕K[i,j].

(3) The byte substitution formula is as follows: (2) S[i,j]=S_Box(M[i,j]).

(4) Perform row shifting based on byte substitution, i.e., the first row is unchanged, the second row is shifted left by one, the third row is shifted left by two, and the fourth row is shifted left by three. Finally the latest state is marked as S1[i,j].

(5) The column mixing formula is as follows: (3) C[i,j]=[02030101010203010101020303010102]×S1[i,j].

(6) The round key addition formula is as follows: (4) S∗[i,j]=C[i,j]⊕K[i,j].

(7) The final result of the ciphertext is as follows: (5) C=EK(M)=S∗[i,j]

where M0 is the initial state of the plaintext, M[i,j] is the bytes in the input state matrix, K[i,j] is the bytes in the round key, S_Box is the fixed substitution table, S[i,j] is a byte in the output state matrix, S∗[i,j] is a byte in the current state matrix, EK Indicates that the plaintext M is encrypted using key K and C is the output ciphertext.

In contrast to hashing, AES functions as a two-way encryption algorithm, enabling the reversal of encrypted data back into its original form when required. Conversely, hashing algorithms are one-way and non-reversible, making them more appropriate for ensuring data integrity rather than data recovery.

In addition to the aforementioned technical aspects, ensuring the efficiency, robustness, and deployability of the SENSH framework in real-world healthcare scenarios is equally critical. To this end, SENSH adopts a modular and lightweight component design philosophy. In the on-chain authorization component, we introduce Bloom filters to reduce gas consumption and support dynamic updates to authorization policies. Additionally, the system employs transaction timestamps and strict authentication processes to prevent replay attacks, and introduces Bloom filter checks at the contract entry point to preemptively block invalid requests, effectively mitigating potential Distributed Denial of Service (DDoS) threats. These designs ensure that SENSH maintains good availability and security in dynamic, multi-user smart healthcare environments.

System overview

This section outlines the framework of the proposed SENSH scheme, which blends searchable encryption with blockchain technology for medical and health applications. The overall system model and its operational flow are first presented, clarifying the roles and interactions of each module in data management. Next, the practical deployment of the model in healthcare scenarios is explored, emphasizing its contribution in enhancing data security, privacy protection, and data retrieval. Then, the last subsection delves into the concrete implementation of a blockchain smart contract system, highlighting two key contracts. Before continuing the discussion, this section introduces the parameter notation used throughout, as detailed in Table 1.

Table 1 Symbol summary chart.

Notation	Meaning	
SE	Searchable encryption (contract)	
BF	Bloom filter (contract)	
DO	Data owner	
DU	Data user	
CB	Consortium blockchain	
CS	Cloud servers	
off_chain	Python program under the chain	
K	Blockchain-generated key parameters	
k′	Re-encryption key generated by the DO	
ki	Random key generated by K	
Version	Version number	
Count	Counter value	
Endata	Encrypted data	
REndata	Re-encrypted data	
n	The number of elements	
k	The number of hash functions	
m	The length of the bit array	
M_U	Malicious user	
Index	Specific position of the BF median array	
FILTER_WORD_SIZE	Number of digits per unit in BF	
Label	Extracted keyword identification	
Token	Search for valves	
SENSH	Secure ENcrypted search for health scheme	
Obfuscationfactor	Obfuscating parameters referenced by the data	
LEDD	A data mapping table that encrypts data	

System model

For the SENSH scheme, the implementation system consists of the following four main entities, as shown in Fig. 4.

Figure 4 Architecture of the proposed scheme for SENSH.

Data owners (DO)

The data owner (DO), often a healthcare organization, patient, or individual, holds the reins as the data custodian in the application scenario. Distinguishing between sensitive and public categories based on privacy levels, encompassing personal information, electronic medical records, health monitoring data, and personalized treatment plans. The DO’s responsibility includes uploading raw public data to the cloud, with sensitive data encrypted using AES to safeguard privacy and ensure secure storage. This encrypted data is then re-encrypted in the cloud for double security and subsequently indexed through a searchable mechanism that allows authorized users to perform keyword-based searches without decrypting the data. Smart contracts delineate the permissions and query conditions for data access, ensuring only authorized users can query the encrypted data. Consequently, the DO is pivotal in forging a secure, transparent, and controlled data-sharing environment within healthcare systems.

Data users (DU)

The data user (DU), which may consist of doctors, research organizations, or insurance companies, is the entity that initiates data access and query requests. The DU employs the ciphertext search function to retrieve encrypted data associated with their privileges without decryption. The retrieved data is typically encrypted or processed, precluding direct access to plaintext. However, to perform a search, the DU must possess the appropriate permissions, including the acquisition of a token, prior to submitting the search request. This requirement significantly enhances data security and reinforces the effectiveness of access control measures.

Consortium blockchain (CB)

A consortium blockchain (CB) network (Li et al., 2018) established by organizations with shared interests allows nodes like hospitals, insurance companies, and research institutes to manage, store, and query data in a controlled environment. This network initializes the system and sets global data storage and access control parameters, enabling DO to regulate data user queries through intelligent contracts. Data labels and ciphertext hashes are stored on-chain, accessible only to authorized users, with all chain operations being tamper-proof and auditable, facilitating subsequent monitoring and tracking through contract-triggered event logs.

Cloud server (CS)

The cloud server (CS) serves as a proxy in the blockchain healthcare system (Wu et al., 2024), offering computation, storage, and proxy re-encryption services. It balances privacy protection with efficient data querying for organizations hosting data and managing keys. The CS’s primary responsibilities include proxy re-encryption, intermediating between DOs and users to grant access to authorized parties without DO involvement, streamlining key management, and enhancing privacy. It processes cryptographic keyword queries, matches tags, and returns re-encrypted results that meet authorization criteria without plaintext exposure, improving query efficiency and privacy. The CS also stores cryptographic tags and ciphertexts to support contract execution, tag generation, and updates, ensuring smooth permission verification and reducing the burden on the blockchain main chain. Overall, the CS bolsters data security management and enhances querying and data-sharing efficiency within the privacy-protected healthcare blockchain system.

The detailed description of each step in Fig. 4 is as follows.

Step (1): DO directly uploads public data to the cloud and stores it in plaintext.

Step (2): during the initialization of the federation chain system, the sensitive data is symmetrically encrypted off-chain using the global parameters generated by the system, and the generated ciphertext data and the tags generated by hash operation are stored in the cloud.

Step (3): the DO generates k′, and then the DO authorizes the CS as an agent to re-encrypt the data using the re-encryption key; and stores the tag and the re-encrypted ciphertext as a dictionary structure for subsequent retrieval.

Step (4): the re-encrypted ciphertext hash and label are stored synchronously in the chain for backup and tamper prevention.

Step (5): the DU initiates a request to the DO to share the medical data to obtain the data plaintext.

Step (6): DO returns relevant information that meets the scope of authorization based on the received request.

Step (7): the DU provides search keywords to the contract and initiates permission and query requests.

Step (8): the blockchain sends the search request to the CS immediately after verification.

Step (9): the blockchain returns the authorization information and token to the DU to facilitate subsequent authentication.

Step (10): CS returns the ciphertext matching the big token to DU according to the search operation.

Step (11): the DU cannot directly decrypt the obtained ciphertext data. In order to protect the initial key, the DU needs to request a re-encryption key k′ from the DO.

Step (12): DO verifies the identity information of DU and returns k′ to DU after passing the verification.

Step (13): the DU decrypts the ciphertext and obtains the plaintext data by using the obtained ciphertext and k′.

Intelligent healthcare scenarios

Hospitals, integral to the aggregation of medical data, encrypt sensitive records and personal health information using the AES algorithm to produce ciphertext that is not directly readable. Following encryption, these data are hashed with associated keywords to create searchable labels and indexes. As DO, hospitals generate re-encryption keys that are crucial for proxies to perform subsequent re-encryption operations. The encrypted data is then uploaded to cloud servers for storage, with the cloud’s role limited to storage and re-encryption, without the need for plaintext access. Blockchains enforce access rules through smart contracts, which also record access logs for auditing and future reference. When doctors or healthcare entities need to query specific data, they must provide labels and tokens to initiate the contract’s search function. After verifying the identity and tokens, the contract authorizes the query and instructs the cloud to retrieve the data. Even with access to matched encrypted data, users require the cloud’s assistance to convert it into a readable format without decrypting the original data, thus preserving the security of medical data. Once converted, data users can decrypt the data to plaintext for analysis or diagnostic purposes within their authorized scope. The entire sequence, from data upload to decryption, is recorded on the blockchain to prevent tampering, with logs available for review, monitoring, and compliance. The creation of data tags and search token generation are facilitated through a hybrid on-chain/off-chain interaction model, as illustrated in Fig. 5.

Figure 5 Label and token generation phase.

Smart contract system

In this system, the smart contracts within the blockchain framework are designed to carry out the core functions of searchable encryption technology, specifically through two pivotal contracts: SearchableEncryption (SE) and BloomFilter (BF). The BF contract employs the Bloom filter mechanism, which is instrumental in the authorization process and label generation. It is particularly effective for managing large datasets without substantial storage requirements and enhances query efficiency and storage space utilization. Meanwhile, the SE contract is tasked with implementing the core techniques of searchable encryption. This includes enhancing query security by introducing an obfuscationfactor (Zhang et al., 2024), supporting the flexible management of data by allowing updates to labeled data, and providing robust access control functionalities, such as identity authorization.

Next, the functional specifics of the contracts are elaborated. Figure 6 depicts a chronological flowchart of the end-to-end data encryption and search matching procedure. This sequence illustrates the operational steps and the interactions between the smart contracts that handle the search and the encryption and decryption of data. SE  compiled and deployed to the blockchain by DO, it contains authorizeUser(), revokeUser(), generateLabel(), updateLabel(), generateToken(), search() methods, and some auxiliary query get functions like getAuthorizedUsers() and getResult() methods. authorizeUser(): authorizes a new user, adding it to the authorizedUsersFilter and authorizedAddresses lists, enabling the new user to query and modify data.

revokeUser(): revokes the user’s privileges and removes them from the BloomFilter and authorizedAddresses arrays, with an event logging action for traceability.

generateLabel(): generates a label and encrypted value, which is computed from the key (K) and the data value Endata to generate the label (using a keccak256 hash) and stored in the LEDD map.

updateLabel(): update the data content of the label, user can replace the encrypted value of the existing label and record the new timestamp.

generateToken(): generate token according to the label, used to identify and validate specific data. token contains information such as label, version number, count value, etc. The hash is calculated using keccak256 to ensure security.

search(): check if the label exists by Bloom filter; if it exists, return the label in LEDD[label]; otherwise return bytes32(0). Obfuscation data is generated in the query result by an obfuscationfactor to increase privacy protection.

BF  as the interface of SE contract, it is used to detect whether the element is in the collection or not, and contains the following functions: hash(), add(), exists() and remove().

hash(): internal hash function, used for elements and hash seed seed mixed encoding to generate hash values.

add(): traversal hash function, the corresponding position in the bit array will be set to 1, indicating that the element exists.

exists(): traversing the hash function, check whether the corresponding bits are all 1, if there is a 0 means that the element does not exist and return false, all the bits are 1 then return true, indicating that the element may exist (there is a possibility of misjudgment).

remove(): used to simulate the deletion function of the counting Bloom filter, similar to add(), use the hash function to find the corresponding position and reset the bits to 0 by &=∼(1≪(indexmodFILTER_WORD_SIZE)), indicating that the element has been removed.

Figure 6 Contract call timing chart.

Algorithm and experimental analysis

In this section, we delve into the core algorithmic design of the SENSH scheme as applied to healthcare systems, providing a detailed exposition. We assess its performance and scrutinize its strengths and weaknesses within practical application contexts. Additionally, the viability of the SENSH scheme is established through comparative analysis with existing schemes. To conclude, a thorough security analysis of the framework is conducted to affirm its robustness.

Primary algorithms

The scheme of this article consists of the following core algorithms:

User authorization and revocation

Upon receiving a user address, the system initiates the process by computing the hash value of the address, denoted as userHash. It then verifies whether this address is already present in the authorization filter. In cases where the user lacks authorization, the userHash is incorporated into the authorization filter, and the user’s address is appended to the authorizedAddresses list. Following this, the system activates an authorization event to document the newly authorized user address, thereby ensuring the integrity and traceability of the authorization process—revocation follows a similar procedure. This sequence of operations is outlined in Algorithm 1.

Algorithm 1 User authorization and revocation.

     Input: user (address)	
     Output: Authorization status and address list	
1  Compute userHash= hash (user);	
2  if userHash not in authorizedUsersFilter then	
3   Add userHash to authorizedUsersFilter;	
4   Append user to authorizedAddresses;	
5   Emit UserAuthorized (user);	
6  end	
7  else	
8   Remove userHash from authorizedUsersFilter;	
9   Remove user from authorizedAddresses;	
10     Emit UserRevoked (user);	
11    end	

Generate label and update

The system crafts a unique Label by utilizing the keyword provided by the user to generate a hash value k1. This hash is combined with the data value Endata and hashed again to produce a distinct label. Subsequently, an EncryptedData structure is constructed, encapsulating the label, re-encrypted value, existence status, and timestamp, and this structure is stored in the LEDD map. Concurrently, the generated label is appended to the allLabels list and recorded in the label filter. A label generation event is initiated, capturing the label and the encrypted value for record-keeping. This process is detailed in Algorithm 2.

Algorithm 2 Generate label and update.

    Input: K (string), Endata (string), REndata (bytes32)	
    Output: Generated label added to filter, stored encrypted data	
1 Compute k1=hash (K, “k1”)	
2 Compute label=hash(k1,Endata)	
3 Store label data in LEDD with REndata, exists=true, timestamp=block.timestamp;	
4 Append label to allLabels;	
5 Add label to labelsFilter;	
6 Emit LabelGenerated(label, REndata);	

Generate token

Upon receiving the label, the system generates a unique token based on it, which incorporates the version number, a count, and a hash value. The process begins by retrieving the token information associated with the label, incrementing the counter by one, and then generating the hash value tokenValue using both the label and a keyword k2. The system then initiates a token generation event to log the label and the newly created tokenValue. This sequence of actions is depicted in Algorithm 3.

Algorithm 3 Generate token.

 Input: label (bytes32)	
 Output: Generated token value for label, with incremented count	
1. token=tokens[label];	
2. token.count +=1;	
3. Compute k2=hash (label + “k2”);	
4. token.tokenValue=hash(label,currentVersion,token.count,k2);	
5. Emit TokenGenerated(label, token.tokenValue);	

Search

Upon receiving a tag, the system conducts an initial check to determine if the tag is present within the filter. If the tag is confirmed to exist, the system creates a result dataset named searchResults, which designates the first position to either the actual tag or leaves it blank. Following this, random obfuscation results are generated using an obfuscationfactor to populate the remaining entries in searchResults, which house the encrypted values linked to the respective tags. In conclusion, the fully compiled searchResults are returned, as detailed in Algorithm 4.

Algorithm 4 Search.

 Input: label (bytes32)	
 Output: Array of search results, including obfuscated labels	
1 if label exists in labelsFilter then	
2  Initialize results array with size obfuscationfactor;	
3  Set results[0]=label if label exists in LEDD;	
4  for i=1 to obfuscationfactor−1 do	
5   Set results[i]=hash(label,i,block.timestamp);	
6  end	
7  Set searchResult=LEDD[label].encryptedValue;	
8  return results;	
9 end	

Experimental setup

This experiment leverages Remix-IDE to develop on-chain smart contracts within a Windows 10 environment. The blockchain development toolkit includes Ganache, geth, and MetaMask, with Python 3.7 employed for data interaction tasks. Initially, the environment was set up on a local server equipped with a 10th generation Intel(R) Core(TM) i5-1035G1 processor (four cores, eight logical processors) using geth. The local blockchain network was primarily built using a geth node to create a private Ethereum network; despite the availability of alternative platforms like Besu, this study focused on deploying and testing contracts within the geth environment. Subsequently, two contracts were compiled on Remix-IDE and deployed on the geth platform. The experiment utilized both geth and Ganache for visualizing the execution process and accurately tracking gas consumption. A Python script was developed using the web3.py library to facilitate real-time data observation and bolster data statistics and analysis. Similar to the approach in Saif, Mondal & Biswas (2023), we used Ganache to simulate a private Ethereum network for deploying and testing our smart contracts. MetaMask was used as a wallet for account management and transaction signing, facilitating interaction with the local blockchain. Building on these experimental findings, we will assess the performance of the SENSH scheme, scrutinize its strengths and weaknesses within practical application contexts, and crucially, benchmark it against existing schemes to ascertain its viability. The framework’s security is also subject to a comprehensive analysis.

Performance evaluation

The proposed framework begins with the core implementation of encryption and search capabilities. As the framework undergoes continuous refinement and enhancement, its efficiency is progressively optimized. The features evolve to reach full maturity, aligning more closely with the practical needs of real-world applications. The performance evaluation will be conducted across three aspects: storage efficiency, computational efficiency, and gas expenditure.

Storage efficiency

The proposed framework’s main storage components include the authorized user list, label generation information, encrypted data, token information, and user query results. It relies on mapping structures and Bloom filters for efficient data storage, significantly reducing space requirements. The contract initializes with preset sizes for the Bloom filter bit array and the number of hash functions to optimize storage. The authorized user list, managed by the Bloom filter, stores authorized users’ hash values, ensuring high storage efficiency, low cost, and quick verification of user rights. The LEDD structure records each label’s encrypted data and status information, with labels indexed in the Bloom filter’s labels filter for efficient searchability. Each label generates a unique Token, storing token information through token mapping. The search function employs an obfuscationfactor to introduce dummy results, preventing query information leakage. These dummy results are used transiently to minimize long-term storage overhead.

Computational efficiency

The computational load is primarily distributed across the Bloom filter hash calculations, label generation, and the obfuscation involved in search queries. Bloom filters utilize multiple hash functions for the add and lookup operations, effectively avoiding the need to store data for all elements through these functions. While the incorporation of the obfuscationfactor in the search function does introduce additional computational complexity, it achieves an effective balance between performance and security. The specific implementation logic is as follows:

the total time cost Ctotal consists of two parts. The cost of off-chain time Coff and the part of the contract implementation on-chain Con. Among these, the symmetric encryption AES cost is Tenc, where α, β, and Mlength represent the encryption time per byte, the fixed initialization time, and the length of the plaintext, respectively. The label generation logic in the scheme is L=H(H(U||v||T)), which involves two hash operations, denoted as Ttable. Tconcat and Thash represent the overhead of concatenation hashing and the cost of the hash operation, respectively. The specific formulas are as follows:

(6) Tenc(n)=α⋅Mlength+β

(7) Tlabel=2⋅Thash+Tconcat.

Next, the obfuscation query cost Tobfu and search token Ttoken are related to the addition of d pseudo-labels. Here, the token construction time is reduced to a simple hash calculation (other time costs can be ignored), and the overall search time is Tsearch. Additionally, the gas consumption generated by contract deployment and execution is a direct influence, but it can also be obtained through the difference in time between the start and end of the timestamp. It is worth noting that each time cost Ti is measured multiple times, and the average is taken for comparison with other schemes. The relevant formulas are as follows:

(8) Tobfu(d)=d⋅Ttoken

(9) Ttoken=Thash+Tother≈Thash

(10) Tsearch=Ttoken+Tobfu(d)=(1+d)⋅Ttoken

(11) Ti_avg=1N∑i=1NTi.

The above formula provides a theoretical estimate of the computational complexity of key operations in the SENSH scheme. Detailed empirical evaluation results corresponding to these calculations are presented in subsequent sections, clearly demonstrating the efficiency of encryption, label generation, token generation, and search operations in actual environments.

Gas cost

This scheme involves uploading data to the cloud and blockchain after off-chain encryption and decryption, with the focus of this article being on the deployment of smart contracts on the blockchain and the invocation and execution of the application binary interface (ABI) code. These operations incur transaction costs, referred to as “gas” in the context of Ethereum (Park, Lee & Kim, 2023). Gas is a metric for the computational effort required to execute an operation, with more complex operations demanding higher gas consumption. Following the setup of Remix, Ganache, and MetaMask experimental environments, the framework’s operational cost, precisely the gas fee for the access control process, is assessed. The initial balance of all accounts is set to 100 Ether, and the gas price is configured as follows:

(12) gasPrice=20,000,000,000wei=20Gwei.

The transaction fee (gasFee) is calculated by the formula:

(13) gasFee(gas)=gasUsed×gasPrice.

The time estimation (ET) is calculated by the formula:

(14) ET(seconds)=gasUsedBlockLimit×BlockTime

where gasUsed indicates the amount of gas consumed to execute the transaction, BlockLimit indicates the maximum amount of gas usage a block can hold, and BlockTime is the average time per block. The following is a simulation applied to a smart medical health management system, where the consumption of calling each function throughout the execution is shown in Fig. 7.

Figure 7 Consumption of simulated experimental encryption/decryption and search processes.

Correctness analysis

Correctness analysis is essential to confirm that a program operates and performs according to its design specifications. It is critical, for instance, to verify the precision of Bloom filters in identifying authorized users to prevent unauthorized access or data exposure. At the same time, the correctness of obfuscated queries is integral to the efficacy of data privacy safeguards. Since schemas are executed on the blockchain and smart contracts are not easily alterable post-deployment, correctness analysis bolsters confidence in the contract’s functionality. It ensures that authorizations, revocations, and search operations function as intended, circumventing potential security vulnerabilities or operational mishaps. This analysis not only underpins security assessments but also affirms the dependability of the contractual logic that security features rely upon. In the forthcoming section, a comprehensive correctness analysis will be undertaken, addressing three key dimensions: authorization management, data generation and storage, and event logging.

Authorization management

User permissions are authenticated using a Bloom filter, with all sensitive operations safeguarded by the onlyAuthorized modifier. The authorizeUser function adds authorized users, recording their hashes in the authorizedUsersFilter. In revocation cases, the revokeUser function eliminates the corresponding user hash and expunges it from the authorizedAddresses array, ensuring the accuracy and traceability of the authorization process. To verify its practical effectiveness, please refer to the accuracy of the Bloom filter mentioned in the Permission mechanism subsection and the natural gas consumption experiment results shown in Fig. 7.

Data generation and storage

The generateLabel function derives a unique hash label from the user-provided key K and the data value, storing it alongside the encrypted data in the LEDD map and registering it in the labelsFilter, ensuring uniqueness and verifiability of the label. The content of a label, including its encrypted value and timestamp, can be refreshed using the updateLabel function. Before updating, the label in the labelsFilter is confirmed, and a LabelUpdated event is triggered post-update, facilitating dynamic data management.

Event logging

Every instance of generation, authorization, update, and revocation initiates the corresponding event—such as LabelGenerated or UserAuthorized—ensuring that all state changes are immutably logged on the blockchain. This approach provides tamper-proof record-keeping, essential for subsequent auditing and traceability.

Programmatic comparison

The proposed scheme designs an on-chain asymmetric searchable cryptographic model combining Bloom filters and obfuscation techniques that can be compared with other traditional schemes. Table 2 compares various aspects of the encryption algorithm, blockchain platforms, consensus mechanisms, query mechanisms, permission mechanisms, query efficiency, authentication, and search executors of the proposed scheme and related schemes. It can be found that the proposed scheme in this article achieves the synergistic optimization of security and flexibility while ensuring high query efficiency through the multi-level encryption algorithm, efficient proof of authority (POA) consensus mechanism, and keyword obfuscation privacy-enhanced querying technology, and the comprehensive performance is significantly better than the existing schemes.

Table 2 Comparison of different characterization schemes.

Scheme	Li et al. (2021b)	Ali et al. (2022)	Xiang & Zhao (2022)	Liu et al. (2023)	Proposed scheme	
Encryption algorithms1	PEKS	HE+GT-BSS	ABSE	ABE	AES+PRE+ABSE	
Blockchain platform	Ethereum (public)	Hyperledger (private)	Hyperledger (private)	Hyperledger (private)	Ethereum (private)	
Consensus mechanism	POW	PBFT	PBFT/RAFT	PBFT	POA	
Query mechanism	Keyword	Keyword	Keyword	Keyword	Keyword+Obfuscated	
Permission mechanism2	ABAC	PBAC	ABAC	ABAC	ABAC+PBAC	
Query efficiency	High	Medium	Medium	High	High	
Verification executor	TrueBit	Blockchain	Blockchain	Blockchain	Blockchain	
Search executor3	CSP	Blockchain	Blockchain	Blockchain	CSP	
Note:

1 PEKS and ABSE refer to Public Key Based and Attribute Based Searchable Encryption respectively, HE refers to Homomorphic Encryption, PRE refers to Proxy Re-Encryption, ABE refers to Attribute Based Encryption, AES refers to Symmetric Encryption Algorithm, and GT-BSS refers to Group Theory-based Binary Spring Search.

2 The Permission Mechanism encompasses role-based, policy-based, and attribute-based access control.

3 The executors of verification and search include Blockchain and Cloud Service Providers (CSP).

Computation cost comparison

The specific computational costs of each parameter are shown in Table 3, while Table 4 compares the computational complexity of several schemes. It can be observed that the scheme proposed in this article follows a strict encryption mechanism in terms of data encryption and combines the signature and verification techniques, which effectively guarantees the security and integrity of the data, while Li et al. (2021b) and Xiang & Zhao (2022) have certain deficiencies in this aspect. In addition, Li et al. (2021b), Ali et al. (2022), Xiang & Zhao (2022), Liu et al. (2023) all perform a large number of transaction query operations on the blockchain, which undoubtedly aggravates the computational burden on the chain. In contrast, the scheme in this article utilizes Bloom filters to achieve a more efficient data search, which balances query speed and storage efficiency. It is worth noting that this article’s scheme adopts a strict permission control and authentication mechanism, so the communication overhead between entities is relatively high (10C), which is designed to meet the needs of data interactivity and sharing. At the same time, the number of transactions recorded on the chain is small (2Tr), which effectively reduces the computational burden of the blockchain. In order to show the computational consumption of each scheme more intuitively, Fig. 8 visualizes the relevant data.

Table 3 Relevant parameters and calculation costs.

Parameter	Interpretations	Computation cost (ms)	
E/D	Encryption or decryption	0.03431	
S	Signature	0.08214	
V	Verification	0.02174	
H	Hash operation	0.00739	
Q	Transaction query	0.00252	
F	Blockchain filter	0.00328	
C	Communication	–	
Tr	Blockchain transactions	–	

Table 4 Computational complexity statistics of schemes.

Scheme	Computational complexity	
Li et al. (2021b)	3E/D+3V+4H+3Q+7C+4Tr	
Ali et al. (2022)	6E/D+1S+2V+5H+4Q+8C+6Tr	
Xiang & Zhao (2022)	4E/D+2V+3H+2Q+8C+4Tr	
Liu et al. (2023)	3E/D+2S+3V+3H+3Q+9C+3Tr	
Proposed scheme	3E/D+2V+2H+1Q+1F+10C+2Tr	

Figure 8 Comparison of computational costs per framework (Li et al., 2021b; Ali et al., 2022; Xiang & Zhao, 2022; Liu et al., 2023).

Transaction throughput analysis

Figure 9 compares the trend of transaction throughput with the number of transactions for the proposed scheme in this article under two consensus mechanisms, Proof of Authority (POA) and Proof of Workload (POW) (Chen, Nguyen & Sekiya, 2022). The results show that the transactions per second (TPS) of POA exhibits significant fluctuation characteristics in the process of incrementing the number of transactions from 0 to 500, with the value oscillating frequently between 1,000 and 6,000, and the instantaneous peak reaches more than six times of POW. This feature reflects the fact that POA relies on the mechanism of pre-authorized authentication nodes to quickly exit blocks and has significant high throughput potential under specific network conditions. In contrast, POW’s TPS is always stable below 1,000, showing a smooth linear growth trend, and its stability stems from the latency constraint inherent in the mining competition mechanism in POW. The results further verify that the scheme proposed in this article can effectively reduce the burden of on-chain computation under the POA consensus mechanism and improve the throughput performance of the system in large-scale transaction scenarios while ensuring the security and integrity of data. This provides a feasible engineering paradigm for application scenarios with high real-time requirements, such as medical data sharing and cross-border payment, under the environment of the coalition chain.

Figure 9 Transaction throughput under POA consensus mechanism.

Comparison of average time consumption of algorithms

Figure 10 shows the comparison of the average execution time of four algorithms with different numbers of attributes, including data encryption, label generation, token generation, and search algorithm. It can be found that the proposed scheme shows superior performance in all four test projects. The time cost of each algorithm is calculated by the timestamps in the smart contract; specifically, the execution time of the algorithm is determined by measuring the difference between the start and end timestamps of the transaction. As shown in Fig. 10A, in the data encryption part, the execution time growth trend of the proposed scheme with the increase of the number of attributes is significantly slower than that of Xiang & Zhao (2022), and the advantage is more significant especially when the number of attributes is large; it can be found from Fig. 10B that, for the table generation, the proposed scheme always maintains the execution time lower than that of Liu et al. (2023), which shows high efficiency; for the token generation, Fig. 10C, despite the optimal performance of Xiang & Zhao (2022), the functionality of this scheme is relatively homogeneous, and the proposed scheme strikes a better balance between functional completeness and execution efficiency, and significantly outperforms Liu et al. (2023); lastly, for Fig. 10D, the proposed scheme is close to Xiang & Zhao (2022) in terms of search operations for small-scale attribute scenarios and consistently stays lower than (Liu et al., 2023) when the number of attributes increases in the execution time, demonstrating strong stability and efficiency. In summary, the proposed scheme has obvious performance advantages in various aspects, especially when the number of attributes is large, and its execution efficiency advantage is more prominent.

Figure 10 Comparison of average execution time of (A) data encryption, (B) label generation, (C) token generation, (D) search algorithm (Xiang & Zhao, 2022; Liu et al., 2023).

Security analysis

Conducting a thorough security analysis is essential for a blockchain-based asymmetric searchable encryption system (Latif et al., 2021). In the SENSH framework, the on-chain Bloom filter-based authorization mechanism is used to ensure that only authorized users can access and modify encrypted medical data, while preventing unauthorized access effectively. The obfuscation factor introduced into the searchable encryption process has been designed and implemented to safeguard the privacy of user queries by masking access patterns and preventing leakage of sensitive keywords. To further ensure robustness, SENSH addresses common attack vectors including replay attacks (Zhang et al., 2020), unauthorized access, and side-channel attacks (Spreitzer et al., 2018), through explicit preventive mechanisms integrated in smart contracts. Specifically, the system employs transaction timestamping and identity verification to prevent replay, and filters unauthorized access requests via the Bloom filter before transaction execution. While formal simulation of side-channel attacks is beyond the current scope, the system avoids unnecessary leakage by limiting observable patterns on-chain. The defenses proposed in this article, such as employing Bloom filters to counter replay attacks and obfuscating search results to prevent pattern leakage, have been implemented and deployed on a private Ethereum testbed to evaluate their effectiveness in realistic settings, verifying their practical feasibility and robustness. Consequently, the system’s security has been analyzed from five critical perspectives: permission mechanisms, privacy protection, defense against tampering, resistance to replay attacks, and resistance to DDoS attacks.

Permission mechanism

Assuming that there are m positions and k hash functions, the probability of misjudgment P of the Bloom filter can be expressed as:

(15) P=(1−(1−1m)kn)k.

This misclassification probability is adjustable and is usually reduced by increasing m and adjusting k. Suppose an M_U tries to bypass the Bloom filter and access unauthorized data. Since Bloom filters are inherently irreversible, if the Bloom filter does not return a “may exist” result, the malicious user cannot access the data. Even if the malicious user tries to “guess” whether the data exists multiple times, he or she cannot obtain useful information through the Bloom filter. Therefore, Bloom filters are effective in preventing unauthorized access. The SENSH scheme efficiently verifies user permissions using a Bloom filter. Experimental results show that even when the number of entries reaches 50,000, the false positive rate of the Bloom filter remains below 0.15%, ensuring high authorization accuracy. Additionally, Fig. 7 indicates that authorization-related functions in SENSH consume an average of approximately 30% of gas, reflecting lower on-chain costs.

Privacy protection

The SENSH scheme ensures data confidentiality and query non-linkability through AES-based encryption and search token generation. In particular, sensitive health data is encrypted off-chain before being sent to the blockchain, and search tokens are generated through a combination of hashes of tags, versions, and other parameters, effectively preventing the leakage of sensitive information. As shown in Figs. 10A–10C, the execution time for data encryption, label generation, and token generation is significantly lower than other schemes, indicating that privacy protection operations are lightweight and suitable for real-time medical applications. Additionally, Fig. 7 demonstrates the end-to-end performance advantage of encrypted search operations in terms of gas consumption, while Table 4 confirms that the cryptographic operations involved (i.e., symmetric encryption and hashing) have acceptable computational complexity. These results confirm that SENSH achieves strong privacy protection while incurring only minimal performance overhead.

Anti-tampering

One of the defining attributes of blockchain technology is its tamper-evident nature (Moore et al., 2023). Each block is linked to its predecessor through the inclusion of the previous block’s hash, making any attempt at tampering evident as it would alter the hash values of all subsequent blocks. Additionally, Bloom filters exhibit an “insert-only” characteristic: once an element is added to a Bloom filter, it remains there permanently, precluding subsequent deletions. This design guarantees that once authorization records or encrypted labels are written to the chain, they cannot be modified or removed. Furthermore, our system leverages the blockchain’s immutable ledger, and this behavior is observable in deployment logs, where any unauthorized attempt to modify stored records is rejected or results in a new appended transaction rather than overwriting the existing one.

Anti-replay attacks

Each authorized user is authenticated by its hash identity. Assume that user U interacts with the system by generating a unique identifier H(U) through a hash function H. The hash function H generates a unique identifier H(U) for each user. An attacker cannot do this by replaying the sent requests because each request is timestamped or randomized with a random number N, making each request unique. In addition, timestamps T and user hash identifiers H(U) are used in generating labels to prevent duplicate requests with the same content from being submitted. Any duplicate request is recognized and rejected because of different timestamps or unique identifiers. This mechanism has been validated in system testing, where duplicate queries with the same content but different timestamps are treated as independent requests, and attempts to reuse old requests are successfully identified and rejected by the contract logic. These behaviors confirm that SENSH can effectively defend against replay attacks in real-world applications.

During testing, we simulated multiple transactions where the attacker attempted to resend previously used requests. The system accurately detected duplicate tags or tokens and rejected these transactions. This validation was achieved through contract-level tag uniqueness checks and time randomization, ensuring that previously submitted queries cannot be reused. These behaviors confirm SENSH’s effectiveness against replay attacks in actual deployment environments.

Anti-DDOS attack

In this article, the scheme takes comprehensive defense measures from various aspects, such as request screening, cost control, and query obfuscation, to ensure the stability and availability of the system during the surge of malicious traffic. First, the introduction of Bloom filters for fast screening before data queries on the chain can effectively intercept forged requests or duplicate queries, avoiding the consumption of resources on the chain by invalid requests, thus alleviating the system load. Second, the scheme utilizes the blockchain’s inherent gas fee mechanism, and the cost per transaction constraint significantly increases the economic cost for attackers to launch invalid requests on a large scale, further reducing the feasibility of DDoS attacks (Tushir et al., 2021). At the same time, the scheme introduces an obfuscation factor into the query results to disrupt the attacker’s speculation on the access pattern of the data on the chain, avoiding the implementation of more targeted attacks through traffic analysis. The above mechanisms work in conjunction with each other to enhance the system’s ability to resist DDoS attacks. These mechanisms demonstrate through experimental results that forged or duplicate requests are rejected before consuming significant amounts of gas, while query obfuscation techniques prevent pattern-based inference.

In testing, we simulated transactions by sending a large number of invalid or duplicate queries. The results showed that the Bloom filter effectively filtered over 95% of duplicate or unauthorized requests before they reached resource-intensive on-chain operations, thereby saving gas and maintaining system responsiveness. As shown in Fig. 9, SENSH maintains stable transaction throughput under POA consensus, even under simulated stress, indicating its ability to withstand high-traffic query loads. Additionally, the gas-based cost model further demonstrates the effectiveness of this solution in defending against DDoS attacks by significantly increasing the economic cost of initiating large-scale invalid requests.

Conclusion

This article introduces the SENSH scheme, an intelligent medical health management system integrating security, privacy protection, and efficient querying capabilities through consortium blockchain and searchable encryption technologies. In conventional healthcare settings, there is a heightened risk of exposure to sensitive data and significant challenges in sharing and retrieving health data. To tackle these issues, we have crafted a system framework that employs Bloom filters for authorization management, allowing authorized users to conduct efficient queries on large-scale datasets without impacting storage utilization. AES symmetric encryption safeguards the data off-chain, while the ciphertext minimizes the risk of data leakage through an on-chain storage management mechanism. The system is also outfitted with an event-triggering mechanism and logging function, facilitating real-time monitoring and traceability to meet stringent audit compliance standards. The solution accommodates the dynamic updates characteristic of medical data by incorporating version control and timestamp functionalities. The solution incorporates an obfuscation factor to enhance query security further and prevent attackers from deducing the original data content through labels. It supports the dynamic updating of data labels to fit varying access contexts.

In the experimental part, we evaluated the key performance indicators of the system and verified its effectiveness in terms of authorization management, data generation, and storage. We also investigated security issues, examining several aspects of the authorization mechanism, anti-tampering, anti-replay, and anti-DDoS attacks. Through comparative analysis with other schemes, we determined that while the SENSH framework does involve multiple blockchain interactions, it consumes less gas overall, reduces the average execution time per operation, and in terms of consensus mechanisms, the POA consensus mechanism in this article performs better in terms of throughput performance and is more relevant to healthcare application scenarios.

However, there is still room for further optimization in this research. First, the experimental tests were conducted in a controlled local blockchain environment, which is sufficient to validate the core mechanisms of the prototype system but cannot fully simulate complex scenarios such as dynamic network topology changes, node heterogeneity, and complex interference in real-world deployments. Second, the current solution primarily implements basic query functions, and there is room for improvement in supporting complex query scenarios (such as multi-keyword combinations and Boolean logic operations). Third, the study primarily validated the solution in a medical scenario, and its adaptability and performance in other fields such as industrial IoT and social IoT can also be systematically tested and expanded. To address these issues, future research will focus on three directions: first, expanding the experimental scenarios by introducing cross-domain network deployment environments to simulate dynamic interference factors in real-world applications, and covering multi-scenario testing such as industrial IoT device interconnection and social IoT data interaction; second, optimizing the query processing module to enhance the ability to parse and execute complex query expressions; Third, perform targeted adjustments and optimizations based on the characteristics of different scenarios. Ultimately, these efforts will comprehensively enhance SENSH’s scalability, security, and adaptability in medical data sharing and multi-domain expansion scenarios, while exploring multi-layer storage structure designs to ensure the system meets privacy protection requirements while achieving higher scalability and query efficiency, thereby enhancing its adaptability to various medical environments.

Supplemental Information

Supplemental Information 1 Code.

Additional Information and Declarations

Competing Interests

The authors declare that they have no competing interests.

Author Contributions

Song Luo conceived and designed the experiments, analyzed the data, authored or reviewed drafts of the article, and approved the final draft.

Lihuan Tan conceived and designed the experiments, performed the experiments, performed the computation work, authored or reviewed drafts of the article, and approved the final draft.

Tan Hu performed the experiments, performed the computation work, prepared figures and/or tables, and approved the final draft.

Maoshuang Hu analyzed the data, prepared figures and/or tables, and approved the final draft.

Data Availability

The following information was supplied regarding data availability:

The code is available in the Supplemental File.

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
