# Peer review of "SENSH: a blockchain-based searchable encrypted data sharing scheme in smart healthcare"

_PeerJ Computer Science, doi:10.7717/peerj-cs.3166_

## Round 0.1 · original submission · Major Revisions

**Language Note:** When you prepare your next revision, please either (i) have a colleague who is proficient in English and familiar with the subject matter review your manuscript, or (ii) contact a professional editing service to review your manuscript. PeerJ can provide language editing services - you can contact us at [email protected] for pricing (be sure to provide your manuscript number and title). – PeerJ Staff

Reviewer 1 ·

Basic reporting

-

Experimental design

-

Validity of the findings

-

Additional comments

This paper presents a searchable encryption-supported, blockchain-based data storage solution for IoT. I have a few comments:

1. Why was AES chosen as the encryption algorithm? Which variant (e.g., AES-128, AES-192, AES-256) has been used?

2. How is the computational cost calculated? Please provide detailed metrics or formulas used in the evaluation.

3. The methodology for setting up and experimenting with the blockchain environment is not clearly explained. More clarity is needed.

4. The author may consider reading the following literature to better understand experimental setup strategies:
"Secure Electronic Health Record Storage and Retrieval Using Blockchain and Encryption for Healthcare Applications".

Reviewer 2 ·

Basic reporting

This paper proposes a comprehensive solution based on consortium blockchain and a searchable encryption data sharing scheme in smart healthcare. The proposed solution, named "SENSH," enables efficient authorization management through Bloom filters and ensures fast queries, and saves storage space. Experimental results show that SENSH offers authorization management, privacy protection, defense against tampering, and anti-replay and DDOS attacks. Compared with existing schemes, SENSH has lower GAS consumption, computation cost, and execution time.

The paper is well-structured and organized, and follows a typical pattern. Relevant details have been provided, and the argument is supported by the experimental data. However, I have a few reservations that need to be addressed:

1- What is the research gap that this research aims to address? A lot of research exists in the field of healthcare using blockchain. How is the proposed SENSH different?

2- How have you ensured that the proposed SENSH is robust and will work in the actual environment?

3- The results are carried out in a controlled environment, which does not necessarily reflect the actual working of SNESH. How have you ensured that the results are not biased?

4- The authors claim that the proposed SNESH is resilient against authorization management, privacy protection, defense against tampering, and anti-replay and DDOS attacks. Which results prove this claim?

5- I see that the proposed SNESH is claimed to be very effective in every aspect. Are there any limitations of SNESH?

Experimental design

-

Validity of the findings

-

---

## Round 0.2 · accepted · Accept

I am pleased to inform you that your work has now been accepted for publication in PeerJ Computer Science.

Please be advised that you cannot add or remove authors or references post-acceptance, regardless of the reviewers' request(s).

Thank you for submitting your work to this journal. I look forward to your continued contributions on behalf of the Editors of PeerJ Computer Science.

With kind regards,

Reviewer 1 ·

Basic reporting

Well revised.

Experimental design

Well revised

Validity of the findings

Sufficient

Reviewer 2 ·

Basic reporting

My comments are addressed, the paper can be accepted.

Experimental design

My comments are addressed, the paper can be accepted.

Validity of the findings

My comments are addressed, the paper can be accepted.

Additional comments

My comments are addressed, the paper can be accepted.